# The Homoleptic Curcumin–Copper Single Crystal (ML_2_): A Long Awaited Breakthrough in the Field of Curcumin Metal Complexes

**DOI:** 10.3390/molecules28166033

**Published:** 2023-08-12

**Authors:** Antonino Arenaza-Corona, Marco A. Obregón-Mendoza, William Meza-Morales, María Teresa Ramírez-Apan, Antonio Nieto-Camacho, Rubén A. Toscano, Leidys L. Pérez-González, Rubén Sánchez-Obregón, Raúl G. Enríquez

**Affiliations:** 1Instituto de Química, Universidad Nacional Autónoma de México, Ciudad de México 04510, Mexico; antoninoarenaza03@gmail.com (A.A.-C.); obregonmendoza@yahoo.com.mx (M.A.O.-M.); mtrapan@yahoo.com.mx (M.T.R.-A.); camanico2015@yahoo.com (A.N.-C.); toscano@unam.mx (R.A.T.); leidyslaura92@gmail.com (L.L.P.-G.); rubens@unam.mx (R.S.-O.); 2Department of Chemical Engineering, University of Puerto Rico-Mayaguez, Mayagüez, PR 00680, USA; willy_meza_morales@hotmail.com

**Keywords:** curcumin–copper (II), single crystal, cytotoxicity, antioxidant

## Abstract

The first single crystal structure of the homoleptic copper (II) ML_2_ complex (M=Cu (II), L = curcumin) was obtained and its structure was elucidated by X-ray diffraction showing a square planar geometry, also confirmed by EPR. The supramolecular arrangement is supported by C-H···O interactions and the solvent (MeOH) plays an important role in stabilizing the crystal packing Crystallinity was additionally assessed by XRD patterns. The log P value of the complex (2.3 ± 0.15) was determined showing the improvement in water solubility. The cytotoxic activity of the complex against six cancer cell lines substantially surpasses that of curcumin itself, and it is particularly selective against leukemia (K562) and human glioblastoma (U251) cell lines, with similar antioxidant activity to BHT. This constitutes the first crystal structure of pristine curcumin complexed with a metal ion.

## 1. Introduction

Curcumin (diferuloylmethane, (1*E*,6*E*)-1,7-*bis*(4-hydroxy-3-methoxyphenyl)-1,6-heptadiene-3,5-dione) [1] is the major components of the Asian spice *Curcuma longa* [2] and has been extensively used in the biologic arena due to multiple purported effects such as antioxidant [3,4,5], anti-inflammatory [6,7], antiviral [8], antibacterial [9,10], antihypertensive, insulin sensitizer [11,12], cytotoxicity against cancer cell lines, and regulation of apoptosis [12,13,14,15,16,17,18,19]. However, this molecule has several disadvantages such as low solubility and therefore, poor bioavailability [20], poor stability, and fast metabolism [21,22], leading to hampered clinical applications [23].

Curcumin consists of an heptanoid chain with a conjugate β-keto-enol system two flanked by two aromatic rings (with an orto-methoxy-phenol system). Different research groups consider phenol groups responsible for their low stability [24] (due to quinoid conjugation type). Thus, the focus has been the synthesis of derivatives introducing alkyl (methoxycurcumin, ethoxycurcumin, butoxycurcumin) or acetyl (diacetylcurcumin, DAC) [25,26]. Additionally, to improve the solubility and overtake the bioavailability disadvantage, the formation of complexes with divalent (e.g., Zn^+2^, Cu^+2^, Mg^+2^) [27,28,29,30] and more scarcely trivalent (e.g., Fe^+3^, In^+3^) metal ions [26,31] has been taken as the principal focus for the synthesis of metal complexes.

Historically, the synthesis of metallic complexes of curcumin began with several metal ions including physiologically relevant ones [27,32] (e.g., Zn, Cu, Mg, Mn, Fe, Se, Ni). However, structural characterizations so far are not conclusive from the solid-state point of view (i.e., Single crystal X-ray diffraction). Therefore, definitive proof of structural evidence of the coordination (ML, ML_2_, or ML_3_) between pristine curcumin and a metal ion was missing until now.

Then, a consistent line of reasoning found in the bibliography is that forming a molecular metal complex consisting of pristine curcumin with a metal ion leads to polymeric arrays due to free phenol groups [33,34]. Wang et al. derivatized the phenol groups using ethoxy or butoxy groups to obtain stable copper (II) ML_2_ crystals [24]. From the biological point of view, Copper (II) is a necessary element to carry out several biological functions and is found widely distributed in connective tissues and blood vessels. A deficiency of this metal can cause different diseases such as anaemia [35], bone fragility [36], and increased susceptibility to infections [37]. In contrast, an excess (intoxication) of copper, is inconvenient and leads to Wilson’s disease [38] and other neurological problems including Alzheimer’s disease [39].

We have placed our efforts in the obtention of the curcumin–copper (II) complex, due to the important properties reported in the literature such as antioxidant, anti-inflammatory, antibacterial [40], antiviral, and antitumoral [41,42]. The chemical study of this complex has been carried out principally using spectroscopic (solution and solid-state) and spectrometric (ionization) techniques [43]. However, an electron density map derived from the X-ray single-crystal structure is now available. Currently, no reports describe the structural elucidation by single-crystal X-ray diffraction (Cambridge Crystallographic Data Center, 2023) between a curcumin molecule and a metal center in the ML_2_ form.

In the obtention of the desired ML_2_ complex of curcumin described herein (that we have named a “cherry on top”), several factors were considered and each one plays its role in turn. Thus, the chelation capability occurs preferentially at the β-diketone system rather than at the phenolic groups which render a stable six-membered ring system; a 2:1 strict ligand-metal stoichiometry. The copper acetate deprotonates the enol function of curcumin and produces a mild acidic media from the resulting acetic acid, which may help prevent the nucleophilic participation of phenol groups. Furthermore, the use of methanol as a solvent was found adequate after several attempts for crystallizing with other solvents.

## 2. Results

We are reporting the first molecular structure of a single crystal of copper and pristine curcumin coordinated in the ML_2_ form. The synthesis of this complex was carried out as depicted in Figure 1.

The solvated (MeOH) structure of curcumin–copper (II) (Complex **1**) crystallized in the triclinic system; space group P-1 (Z = 1) is shown in Figure 1 and the crystallographic data are shown in Appendix A. The curcumin (1E,6E)-1,7-bis(4-hydroxy-3-methoxyphenyl)-1,6-heptadiene-3,5-dione), also known with the common name diferuloylmethane, exhibits keto–enol tautomerism in solution (MeOH) adopting the enol tautomeric form (1E,4Z,6E)-5-hydroxy-1,7-bis(4-hydroxy-3-methoxyphenyl)hepta-1,4,6-trien-3-one (Figure 1). Assuming this tautomeric form, curcumin still has four different rotation axes that can give rise to conformational isomers three of which have been determined by single crystal X-ray diffraction [44,45] (CSC refcodes: BINMEQ, BINMEQ12 and QUMDIN). The anti, s-cis, s-trans, anti-conformation (QUMDIN), where the term anti indicates the orientations of OCH_3_ groups referred to the keto–enol moiety, was found for the ligand in its deprotonated form. Although the ligand molecule is nearly planar, the orientations of the phenyl groups are slightly twisted [C6-C7-C14-C15 = −2.1(3)° and C2-C1-C8-C9 = 8.4(4)°].

In the homoleptic complex (**1**), the copper atom resides on an inversion center with a nearly perfect square planar (τ_4_ = 0.00) coordination. The four coordination sites are occupied by oxygen-donor atoms of the 1,6-heptadiene-3,5-dione moieties of curcumin. Thus, each curcumin acts as a bidentate chelate ligand Figure 1. The corresponding bond lengths are Cu-O1 = 1.9191(15) Å and Cu-O2 = 1.9192(14) Å, O1···O2 = 2.807(2) Å and O1···O2(−x, 1 − y, 1 − z) = 2.618(2) Å. The deviation of the mean planes of ligands to the coordination plane is 11.0°. Selected bond lengths are summarized in Table 1.

The metal complex (**1**) and pristine curcumin were screened against six cancer cell lines (see Appendix A), and the metal complex was rendered more potent than curcumin itself, with selectivity against leukemia and glial cancer lines. The IC_50_ values were obtained for these two cell lines. The antioxidant tests were recorded in the lipid peroxidation inhibition model (TBARS), showing that the metal complex of curcumin acts against lipid oxidation.

## 3. Discussion

During the synthesis of the curcumin–copper complex (**1**), our main goal was the obtention of a single crystal of ML_2_ type. Both solvent and raw materials were of high purity and a rigorous stoichiometry was used. Additionally, the filtrate and the solid precipitate were kept during the process for further workup. Further analysis (see ESI in Appendix A) showed that both solid and filtrated liquid contained curcuminoid fragments and metal.

However, the precipitate was found barely soluble even in large amounts of methanol and rendered an amorphous solid after full evaporation. We attribute this behavior to the possible polymeric nature of the product (m.p. = 290°). On the other hand, the filtrated mother liquor (containing metal-complex and a mixture of acetic acid/methanol) tends to spontaneously form crystals after slow evaporation and concomitant cooling, and these crystals redissolve readily in methanol at room temperature in an approximately 1:10 *w*/*v* ratio. In the latter case, the formation of a single crystal (suitable for X-ray diffraction) was successfully achieved by slow evaporation (see pictures in Appendix A).

EPR studies of crystals and amorphous solids (Figure 2 and Appendix A) show a typical four-line pattern of paramagnetic copper nuclei. The g*‖*, g⊥, A*‖*, and A⊥ values were obtained directly from the EPR spectra. The g*‖* and g⊥ values of complex **1** were 2.290 and 2.059, resulting from unpaired electrons in the *d*x^2^-y^2^ molecular orbital [46]. The value of g*‖* is <2.3, suggesting a covalent environment for this complex and the A*‖* = 164 G supports the square-planar coordination of 4 equivalent oxygen atoms around the copper (II) ion. These parameters agree with those previously reported [28,47,48]. Furthermore, the CP/MAS spectrum of the ligand curcumin shows good resolution, contrasting with the very broad signals of the paramagnetic complex **1** (Appendix A).

Comparison of the IR-ATR spectra were carried out between (a) ligand and complex **1** and (b) amorphous material and monocrystals. The IR of curcumin (see Appendix A) shows signals at 3499 cm^−1^, attributed to phenols (Ar-OH); stretching signals for the carbonyl group (C=O) at 1626 cm^−1^, and the signal observed in 1601 cm^−1^ corresponds to vinyl groups (C=C). It has been reported that the intense IR signal at 1504 cm^−1^ is due to a mixture of vibrations of ν(C=O), δ(C-C=C), δ(C-C=O), and aromatics at ν(C=C), ν(C=C-H) [49,50]. Complex **1** gives characteristic signals for the phenolic groups at 3443 cm^−1^; the characteristic signals to the carbonyl group ν(C=O, 1619 cm^−1^) and the vinyl groups (C=C, 1587 cm^−1^), as well as the intense signal of 1491 cm^−1^, are red-shifted with respect to the ligand due to Cu (II) coordination, as previously reported [49]. The IR spectrum of complex **1** shows a new signal at 478 cm^−1^ due to copper bonding to oxygen [8].

The IR spectra of both materials (amorphous and monocrystals, see Appendix A) show minimal but subtle differences, observed in the region of the phenol groups (amorphous 3441 cm^−1^ (broad signal) and crystals 3443 cm^−1^ (less broadened)). The additional signal at 3537 cm^−1^ for the crystalline complex is assigned to solvent molecules (methanol).

The UV-Vis spectrum of curcumin (1 µM) exhibits a maximum absorption peak at 420 nm in methanol (see Appendix A) and is attributed to π → π * transitions [51]. The UV-Vis spectrum of complex **1** (1 µM) shows two characteristic absorption bands: 421 nm and 435 nm in methanol (see Appendix A) corresponding to the strong complexation between curcumin and copper (II) [51], these 2 absorption bands have also been attributed to charge transfer phenomena from curcumin to copper (II) [43,47].

The XRD pattern of complex **1** (single crystals) showed characteristic peaks (3.1, 3.5, 3.9, 4.3, 6.0, 7.1, 8.8, 12.1°), reflecting its crystallinity. However, the product that is retained in the filter shows a clear pattern of amorphous material, while the peaks found at 3.1 and 8.7° correspond to residual crystalline material retained (See the overlapping patterns in Figure 3 and individual patterns in Appendix A).

It is evident from PXRD patterns (Figure 3) that the material retained in the filter is almost completely amorphous with low crystallinity, while complex **1** displays good crystallinity. Despite the different temperatures between the simulated (130 K) and the experimental (298 K) patterns, enough correlation between them is observed, with noteworthy persistence of the 2,−1,2 diffraction peaks which can be associated with the supramolecular structure of complex **1** (See Appendix A including calculated pattern from obtained crystals).

In the crystal structure, the solvent molecules of methanol play key a role in the crystal architecture linking two neighboring complexes through two strong O4–H4···O8 and O8–H8···O6 hydrogen bonds generating a centrosymmetric cyclic dimer [graph set descriptor: $R_{4}^{4}(44)$] which extends infinitely. Furthermore, each 1D infinite chain interacts with two adjacent chains by non-classical hydrogen bonds (C21–H21A···O3 and C21–H21A···O4) forming a 2D sheet parallel to the (2,−1,2) crystallographic plane, Figure 4a. These sheets are interlaced by hydrogen bonds with the second methanol solvent molecule C22–O7 acting as a donor (O7–H7···O2) and acceptor (O6–H6···O7), generating another centrosymmetric cyclic dimer [graph set descriptor: $R_{4}^{4}(28)$] associated to an aromatic ··· metal chelate stacking Figure 4b.

The third methanol solvent molecule was found to be disordered around an inversion center and plays the role of void filling, although some contacts can be identified (Table 2).

To obtain further information about close contacts in the crystal packing, the Hirshfeld surfaces were mapped using Crystal Explorer 21.5 v software [52,53]. The surface areas mapped with the shape-index (Figure 5) and the electrostatic potential (Figure 6) functions for the complex revealed close contacts between the phenolic ring and the metal chelate. The blue bump shape belongs to the donor, and the red hollowed spots correspond to the acceptor in an intermolecular interaction.

The surface area was determined by the d_norm_ function for complex **1**, revealing close contacts (red regions) on the phenolic and methoxy groups (Figure 7). However, the 2D fingerprint plot shows the characteristic two peaks corresponding to O···H/H···O contacts (Figure 8a). Also, the greater percentage of contacts originate from H/H interactions followed by O···H/H···O contacts with 44.2 and 24.5 percent, respectively (Figure 8b).

A survey for curcumin transition metal complexes on the Cambridge Structural Database, CSD version 5.42 updates (April 2023) [54] shows 23 structures where all of them contain just 1 molecule of curcumin as a ligand. Of these 23 structures, 8 [55,56,57,58,59,60,61,62] (CSD refcodes: PIRVET, HUKGEC, GEGLAH, FEGCOL, BUBSOJ (two molecules), DEYLUR, DOQWUE, GESYAG) display the *anti*, *s-cis*, *s-trans*, *anti*-conformation for the ligand. A comparison among this subset, including the curcumin ligand (QUMDIN) and complex **1**, was performed using the program CrystalCMP [63]. On the calculation of the differences in interatomic distances and the similarity is calculated as a positional difference between molecules in a representative molecular cluster. A very good agreement was found between the structure of ligand in Complex **1** and the corresponding structures found in the subset. A lesser agreement was observed for the pure ligand structure and that found in the complexes (Figure 9).

Complex **1** was four times more active against the U-251 cell line (see Table 3) than the ligand (curcumin), which agrees with previous reports [64] and suggests that: (a) curcumin and the metal ion are mutually stabilized in the complex [27,65,66], (b) solubility and bioavailability are improved [67], and (c) the subsequent cell death occurs as a consequence of intercalation with DNA [68].

The log *p* value of curcumin is 3.1 ± 0.04, consistent with previous reports [70,71], while the log *p* of complex **1** was 2.3 ± 0.15, indicating that the latter has better water solubility and therefore better bioavailability.

The antioxidant assay (TBARS) of the copper–complex (**1**) showed enhanced activity with respect to the ligand (almost 3 times, see Table 4) and practically equaled that of BHT (Butylated hydroxytoluene), demonstrating that the copper within the complex plays an important role in the inhibition levels of malondialdehyde (MDA). Also, it is known that the curcumin–Cu^+2^ complex enhances the activities of several antioxidant enzymes (catalase, superoxide dismutase, and glutathione peroxidase) and attenuates the rise of MDA levels [72,73].

There is a concrete benefit in knowing the precise molecular structure from the X-rays determination of the single crystal of curcumin with copper since it will help to solve structural unknowns (geometry or metal-ligand relationship), as well as to explain different biological effects found, i.e., the antioxidant capacity of the complex, the high cytotoxic potential against cancer cells, and the potential of copper curcumin complex to combat Alzheimer’s disease. Furthermore, it will support studies and scope in computational chemistry (QSAR, DFT, molecular docking, etc.).

## 4. Materials and Methods

Anhydrous copper (II) acetate was available commercially and purchased from Sigma-Aldrich. Pure curcumin was obtained from synthesis as previously reported [74]. The solvents were HPLC grade from Sigma-Aldrich.

### 4.1. Physical Measurements

The melting points were determined on an Electrothermal Engineering IA9100 *×* 1 melting point apparatus and are uncorrected.

### 4.2. Spectroscopic Determinations

The IR-ATR absorption spectra were recorded in the 4000–400 cm^−1^ range on a FT-IR NICOLET IS-50, Thermo Fisher Scientific spectrophotometer. The EPR spectra were recorded in DMSO at liquid nitrogen temperature (77 K) on an Electron Paramagnetic Resonance Spectrometer JEOL, JES-TE300, ITC Cryogenic System, Oxford. Mass spectra were recorded in a Bruker Esquire 6000 with electrospray, atmospheric pressure chemical ionization, and ion trap (ESI-TI). Uv-Vis was recorded on UV-VIS SHIMADZU 1800. CP/MASS ^13^C NMR spectra were recorded on a Jeol 600 MHz spectrometer (15.0 kHz of MAS) using adamantane as the reference (298 K).

### 4.3. Biological Assays

The cytotoxicity of curcumin and curcumin–copper complex **1** was tested against 6 cancer cell lines: U251 (human glioblastoma cell line), PC-3 (human Caucasian prostate adenocarcinoma), K562 (human Caucasian chronic myelogenous leukaemia), HCT-15 (human colon adenocarcinoma), MCF-7 (human mammary adenocarcinoma), and SKLU-1 (human lung adenocarcinoma). The cell viability in the experiments exceeded 95%, as determined with trypan blue. The human tumor cytotoxicity was determined using the protein-binding dye sulforhodamine B (SRB) in a microculture assay to measure cell growth, as described in the protocols established by the NCI and previously [69]. A dose–response curve was plotted for each complex and the concentration (IC_50_), resulting in an inhibition of 50% estimated through non-linear regression analysis. Results were expressed as inhibitory concentration 50 (IC_50_) values.

Antioxidant activity of curcumin and curcumin–copper complex were tested by Thiobarbituric Acid Reactive Substances (TBARS) method and were measured using rat brain homogenates according to the method described by Ng and co-workers [75], with some modifications as previously was recorded [76]. The concentration of TBARS was calculated by interpolation on a standard curve of tetra-methoxypropane (TMP) as a precursor of MDA [77]. Results are expressed as n moles of TBARS per mg of protein. The inhibition ratio (IR [%]) was calculated using the formula IR = (C–E) × 100/C, where C is the control absorbance and E is the sample absorbance. Butylated hydroxytoluene (BHT) was used as a positive standard. All data are presented as mean ± standard error (SEM). Data were analyzed by one-way analysis of variance (ANOVA) followed by Dunnett’s test for comparison against control. Values of *p* ≤ 0.05 (*) and *p* ≤ 0.01 (**) were considered statistically significant.

### 4.4. Determination of Partition Coefficients

Partition coefficients were measured according to the shake–flask method previously reported [78,79], and the quantitation of the curcumin (8 μM) and Complex **1** (4 μM) was recorded by ultraviolet (UV) spectrophotometry. The experiments were conducted in triplicate and standard curves are found in the Appendix A.

### 4.5. Synthesis of Complex ***1***

In a 100 mL ball flask with 30 mL of methanol-HPLC, 200 mg of curcumin (0.54 mmol) was added and heated, and a methanolic (50 mL) solution containing 50 mg of anhydrous copper (II) acetate (0.27 mmol) was added dropwise. The reaction mixture was refluxed for 12 h. The volume was reduced to approximately 20 mL and the suspension was filtered off in a Hirsch funnel; the solid retained in the filter was washed with cold methanol water (8:2 *v*/*v*) and was left to stand at *vacuum* and room temperature for 4 hours. Solid phase: Amorphous solid, dark brown color, Yield= 20%. Melting point: 290–292 °C. MS (ESI^+^) = 797.0 *m*/*z* [M]^+^, 900.9 *m*/*z*. IR-ATR 3441 (O-H, phenol), 2999-2833 (C-H_Ar_) 1619 (C=O), 1587 (C=C), 1491 (C=O-Cu), 965 (C–O, β-diketone), and 469 cm^−1^ (O-Cu).

The filtrate was slowly evaporated, rendering crystals that were filtered off in a Hirsch funnel and washed with cold methanol water (8:2 *v*/*v*). Crystals were left to stand in a *vacuum* at room temperature for four hours. Crystals, bright brown color, Yield= 50%. Melting point: 230 °C (decomposition). MS (ESI^+^) = 820.0 *m*/*z* [M + Na]^+^, 898.0 *m*/*z* [[M + Na + DMSO]^+^. IR-ATR 3587 (O-H, MeOH), 3443 (O-H, phenol), 3039-3837 (C-H_Ar_), 1621 (C=O), 1587 (C=C), 1496 (C=O-Cu), 963 (C–O, β-diketone), and 478 cm^−1^ (O-Cu). Elemental Analysis: Calculated for C_42_H_38_CuO_12_·0.5H_2_O: C 62.49; H 4.87; experimental: C 62.32; H 4.80.

Single crystals formation (suitable for X-ray diffraction): 30 mg of crystals described previously were dissolved in 30 mL of anhydrous methanol-HPLC and filtered for elimination of solid impurities, this solution afforded crystals after slow evaporation for one week at room temperature.

### 4.6. Powder and Single-Crystal X-ray Diffraction Analysis

Powder X-ray diffraction (PXRD) was performed in the transmission mode on a BRUKER D8 ADVANCE diffractometer, under Cu Kα radiation (λ = 1.54059 Å). The equipment was operated at 40 kV and 40 mA, and data were collected at room temperature in the range of 2*θ* = 3–70°. PXRD data analyses were conducted using WinPLOTR [80].

Single-crystal X-ray diffraction studies for complex **1** were carried out on a Rigaku Xcalibur and Gemini with MoKα radiation (λ = 0.71073 Å) by performing ω scans frames at 130 K. Absorption correction was carried out empirically using SCALE3 ABSPACK scaling algorithm (CrysAlisPro, Agilent Technologies, Version 1.171.36.32 [81]) and refined by full-matrix least-squares treatment against |F|2 in anisotropic approximation with SHELXL-2019/3 [82] in the ShelXle program [83]. H-atoms were included in the geometrically calculated positions. Visualization and analysis of crystal structures were carried out using Mercury 2021.3.0 v [84] software.

## 5. Conclusions

We have successfully achieved the synthesis and the complete structure elucidation of the homoleptic ML_2_ curcumin–copper complex, authenticated through single crystal X-ray diffraction, unequivocally demonstrating the 2:1 curcumin–metal ratio. An outstanding feature is that no substitution of the phenolic groups was necessary.

The supramolecular analysis demonstrates that the solvent plays a critical role in crystallization. Furthermore, the mother liquor (filtrate) should be considered a suitable source of crystals and not be underestimated. The supramolecular interactions in both amorphous and crystal phases are specific and different, with the solvent of crystallization playing a critical role. These subtle differences may be the key factor that has prevented the crystallization of pristine curcumin with a single-atom metal in the past.

The biological activity of the homoleptic curcumin–metal complex shows a dramatic increase in cytotoxicity and antioxidant properties compared to pristine curcumin. Therefore, the search for other homoleptic crystal structures of curcumin with different metals is a task to be undertaken in the short future.

Therapeutic areas such as Alzheimer’s disease, degenerative neurological ailments, and cancer research, and the search for antioxidant agents may hopefully expand their scope from the present finding.

## Data Availability

Not applicable.

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
