# Peer review of "The Homoleptic Curcumin–Copper Single Crystal (ML2): A Long Awaited Breakthrough in the Field of Curcumin Metal Complexes"

_molecules, 2023, doi:10.3390/molecules28166033_

Round 1

Reviewer 1 Report

The authors describe the synthesis and characterization of a Cu(II) complex based on curcumin ligand. The resulting complex was extensively characterized by single crystal X-ray diffraction and studied againt cancerous cell lines. The manuscript presents some interesting results and findings. However, prior to publication, I would like to suggest the authors revise the manuscript in the following aspects:

1.      There is hardly any discussion on UV-Vis and EPR spectra. The authors should expand the discussion, especially in the characterization section.

2.      Provide overlapped UV-Vis spectra of the ligand and complex. Also include the conditions.

3.      Provide an explanation for ALERT level B.

4.      Section 4.4: Provide mmols of the ligand and metal. Replace Celsius with °C, beta with β, etc.

5.      Include IR data for the missing functional groups such as C=C, C-O, C-H(al/ar), etc.

6.      I cannot see NMR (1H and 13C) data in the manuscript. Including NMR discussion and spectra to enhance the quality of the paper and help researchers working in this area.

7.      One of the inherent problems with curcumin is its low solubility. Authors should compare and include a discussion on the solubility of the ligand and complex.

8.      Expand discussion related to biological studies.

The overall quality of the English is low and should be revised to improve the overall readability of the manuscript.

Author Response

Please read the attached PDF document.

Reviewer 2 Report

The authors focused on the preparation of a single crystal structure of the CuL2 complex, where L is curcumin. The cytotoxic and antioxidant activities of the complex were also investigated and compared with those of the ligand.

The Cu(II) complex with curcumin was found to be more active than the ligand against human tumor cell lines U-251 and K-562. Furthermore, this complex shows enhanced antioxidant activity towards the ligand.

A large number of metal complexes with curcumin have been synthesized and characterized. A copper complex has also been synthesized, but a complex with chemical structure ML2, proven by single crystal X-ray diffraction, has not been obtained and published.

The authors proved the structure of the Cu(II) complex not only by single crystal X-ray diffraction but also by elemental analyses, melting point and IR spectra. What about NMR spectra? NMR spectra of the complex compared to the ligand (curcumin) will be very important for the biological activity of the metal complex. It will be very interesting to see if the structure in the solid state will be exactly the same as that in solution.

Furthermore in section 2 “Results”, the authors have explained in great detail the crystal structure of the complex and have not paid any attention to the IR spectra and other studies proving the structure of the complex both in solid and liquid state. In this regard, I recommend that the authors do the NMR spectra and discuss them.

Minor editing of English language required.

Author Response

Please read the attached PDF document.

Reviewer 3 Report

In this paper, Arenaza-Corona et al. present single crystal growth of the homoleptic copper (II) ML2 complex and its structural analysis. This paper is written well and will be interesting for the reader.

According to my opinion, this paper must include the following points before the acceptance:

1) The title must be changed such as “Single crystal growth of the Homoleptic Curcumin-Copper (ML2) or Growth of the Homoleptic Curcumin-Copper Single Crystal (ML2).

2) The author should present xrd patterns as a figure in the main paper.

3) EPI and EPR spectra are shown in the supplementary materials. I would suggest, the author should depict some of the spectra in the main manuscript.  

Author Response

Please read the attached PDF document

Reviewer 4 Report

The manuscript “The Homoleptic Curcumin-Copper Single Crystal (ML2): A long awaited cherry on the cake in the field of curcumin metal complexes” by Antonino Arenaza Corona, Marco A. Obregón Mendoza, William Meza Morales, María Teresa Ramírez Apan, Antonio Nieto Camacho, Rubén A. Toscano, Leidys L. Pérez González, Rubén Sánchez Obregón, Raúl G. Enríquez concerns the synthesis and studies of the bischelate copper(II) complex of curcumin. The authors were lucky to obtain the letter in the single crystal form and to solve for the first time the crystal structure of curcumin bischelate complex of Cu(II), which cytotoxic and antioxidant activities have also been studied. In general, the article is well written and compiled. However, in its present form, the manuscript cannot be published, as there are significant shortcomings that require correction.

The experimental part differs from the rest of the text in the quality of the English, which needs considerable improvement. Also in the Discussion section there is no comparison of spectroscopic data for a sparingly soluble precipitate and a bischelate crystalline complex.

It should be noted that from our experimental practice, prolonged boiling of the components promotes the formation of polymer complexes.

It would be extremely useful to place in the SI IR the free curcumin spectrum. In addition, powder diffraction patterns of both solid products described in the synthesis section are needed to characterize the phase purity of the complex 1.

Line 73. The authors claim that “the use of a mild acidic media afforded by the copper 73 acetate was used to avoid nucleophilic participation of phenol groups” but copper acetate is generally used for the ligand deprotonation.

Line 129 How was the absence of crystallinity proven?

Data for CHN-analysis are more consistent with the formula C42H38CuO12(H2O)0.5

Remove the check-cif information from SI. Upload a new check-cif file to CCDC including a comment on alert B.

Move the tables with geometric parameters of the structure of 1 to SI.

Line 211. Please decipher the abbreviation RMSD.

The authors have placed UV-VIS spectra in SI, but do not comment on them in the main text of the article.

In the captions for Figures S8 and S9, change spectrA to spectrUM.

In conclusion, I would like to note that the drawings made on a black background hurt the eyes.

Comments and minor corrections that are not included in the current list are made directly in the PDF file of the main text of the manuscript, which is included in the review.

The experimental part differs from the rest of the text in the quality of the English, which needs considerable improvement. 

Author Response

Please read the attached PDF document.

Round 2

Reviewer 1 Report

The authors have revised their MS satisfactorily.